# Authorship trends in infectious diseases society of America affiliated journal articles conducted in low-income countries, 1998–2018

**Chelsea E. Modlin**[1]*, **Qiao Deng**[2], **David Benkeser**[2], **Yimtubezinash Woldeamanuel Mulate**[3], **Abraham Aseffa**[4], **Lance Waller**[2], **Kimberly R. Powell**[5], **Russell R. Kempker**[6]*

**1** Department of Medicine, Division of Infectious Diseases, Johns Hopkins University School of Medicine, Baltimore, Maryland, United States of America, **2** Department of Biostatistics and Bioinformatics, Rollins School of Public Health, Emory University, Atlanta, Georgia, United States of America, **3** Department of Microbiology, Immunology, and Parasitology, Addis Ababa University School of Medicine, Addis Ababa, Ethiopia, **4** Armauer Hansen Research Institute, Addis Ababa, Ethiopia, **5** Emory University Woodruff Health Sciences Center Library, Atlanta, Georgia, United States of America, **6** Department of Medicine, Division of Infectious Diseases, Emory University School of Medicine, Atlanta, Georgia, United States of America

\* cmodlin2@jh.edu (CEM); rkempke@emory.edu (RRK)

**Data Availability Statement:** The authors confirm that the data supporting the findings of this study

## Abstract

An increasing amount of infectious diseases research is conducted in low-income countries (LIC) given their high burden of disease; however, the contribution of LIC investigators as measured by authorship metrics, specifically to infectious diseases research, has not been thoroughly studied. We performed a literature search for primary research conducted either within LICs or using samples from LIC participants published between 1998–2017 in the Infectious Disease Society of America-affiliated journals *Clinical Infectious Diseases*, *Journal of Infectious Diseases*, and *Open Forum Infectious Diseases*. Primary outcomes included proportion of LIC-affiliated first and last authors (i.e. lead authors) per year and authorship trends over time. Secondary outcomes included proportion of LIC-affiliated authorship by geographic distribution and disease focus. Among 1308 publications identified, 50% had either a first or last LIC-affiliated author. Among these authors, 48% of LIC-affiliated first authors and 52% of LIC-affiliated last authors also reported a non-LIC institutional affiliation. While the absolute number of articles by LIC-affiliated lead authors increased over the 20-year period, the proportion of articles with LIC-affiliated lead authors decreased. There is a growing literature for infectious disease research conducted in LICs yet authorship trends in a small subset of these publications demonstrate a pronounced and worsening exclusion of LIC-affiliated investigators from publishing as lead authors.

are available within the article and its supplementary materials.

**Funding:** This work was funded by Imagine, Innovate and Impact (I3) Funds from the Emory School of Medicine and through Georgia CTSA NIH award (UL1-TR002378). C.E.M. was supported as a post-doctoral fellow of the Oxford-Johns Hopkins Global Infectious Disease Ethics Collaborative funded by the Wellcome Trust [grant numbers 221719 and 216355]. The funders had no role in study design, data collection and analysis, decision to publish, or preparation of the manuscript.

**Competing interests:** The authors have no competing interests.

## Introduction

Morbidity and mortality from infectious diseases disproportionately affects developing parts of the world. Within low-income countries (LICs), almost one-third of deaths are caused by lower respiratory infections, HIV/AIDS, diarrheal illness, malaria or tuberculosis [1]. These conditions were specifically included as one of the eight goals within the United Nations Millennium Development Goals [2], revised in 2015 as the Sustainable Development Goals [3], indicating the commitment of international governance to address these issues within low-income settings. While the predominant burden of infectious diseases falls within LICs, these are the geographic areas that have historically—and continue—to face significant systematic barriers to achieving equitable academic resources, institutional infrastructure, and economic development compared to high-income countries (HICs) [4, 5]. Although there have been escalating calls for developing research capacity within LICs [6–8], the establishing of research priorities, definitions of what constitutes 'meaningful' research, and financial support for both primary research and research capacity building are dominated by HIC voices. Combined with the greater academic, institutional, and economic resources at the disposal of HIC investigators these inherent asymmetries in global health and infectious disease research end up perpetuating existing academic inequalities. Publication authorship plays an important role as a measure of local research capacity [9]. Authorship denotes leadership within the research team and significant contribution to the concept, design, and conduct of research [10]. At an individual level, authorship can directly impact academic tenure and grant funding. Retention of these investigators within the LIC academic infrastructure feeds back into the sustainability of future research production [11, 12].

To better understand research capacity and authorship equity in infectious diseases, we systematically reviewed publications in three infectious disease journals that were conducted in a LIC research setting or enrolled LIC participants. The main objective of our study was to analyze trends in authorship affiliation within infectious disease research over the past two decades and explore how this has changed over time with growing attention toward global health equity [13, 14]. Our goal is to highlight recent publication trends of LIC investigators and inform future efforts that promote LIC authorship equity and sustainable research capacity.

## Materials and methods

### Search strategy and data extraction

The focus of this study was primary research–defined as data collection, human subject research, or research using specimens from human subjects—located in one or more LICs and published in one of the Infectious Diseases Society of America (IDSA) journals. The IDSA journals (*Clinical Infectious Diseases*, *Journal of Infectious Diseases*, and *Open Forum Infectious Diseases*) were selected due to the society's record of publishing high-impact research, influence on the establishment of global standards for clinical care, and commitment to developing global health partnerships and furthering the international physician-scientist workforce [15]. This study was declared exempt from review by the Emory University Medicine Institutional Review Board.

We conducted an electronic literature search to locate relevant studies using Scopus abstract and citation database from Elsevier publishing. To distinguish these journals from those with similar root titles, IDSA journals were identified by print and electronic ISSN number(s): *Journal of Infectious Diseases* (0022–1899; 1537–6613), *Clinical Infectious Diseases* (1058–4838; 1537–6613), *Open Forum Infectious Diseases* (2328–8957). Publication records

were limited to the 1998–2017 period. Of note, Open Forum Infectious Diseases journal first published in 2014. Results were then explored using an algorithmic text and keyword search in Microsoft Excel for associations with LICs based on title, keywords or abstract text. If no association was identified by this method or if multiple countries were identified, a full manual review of the text was undertaken to identify study locations. For each record with LIC settings or participants, the Scopus citation, bibliographical, abstract and keyword information were extracted.

## Inclusion and exclusion criteria

Studies were included with the following criteria: (1) published in an IDSA journal between the period of January 01, 1998 through December 31, 2017, (2) conducted in one or more LICs as defined by the World Bank [16] according to year of publication. A study was excluded if the format was not primary research, did not involve human subjects, or was a multisite international study that included both LIC and non-LIC settings or subjects. Excluded article types included systematic reviews, meta-analyses, perspectives, commentaries, notes, news articles, and quizzes. Publications were also excluded if the primary location of the study was not disclosed.

## Analysis

Citations were stratified by year of publication. The abstract, title, keywords and authorship affiliations were cross referenced with LICs for that year as classified by the World Bank to determine LIC affiliation of the study location and authors. Data were censored in years in which a country was no longer classified as low-income. For authors reporting dual affiliation in both LIC and non-LIC countries, these were included as LIC-affiliated authorship. A subanalysis was conducted to investigate any differences in trends between LIC-only authors and authors reporting dual affiliation. International geographic regions were identified as defined by the United Nations [17]. For international studies that included multiple LIC sites, the publication counted once towards each geographic region involved. Similarly, for studies with a focus on more than one disease, the publication counted once towards each disease studied.

We summarized trends over time by computing the proportion of papers published in each year with a first or last author from a LIC. The linear trend in these proportions over time was summarized using simple linear regression. We calculated adjusted associations between study characteristics and the probability of having a first or last author from a LIC using logistic regression and standardization approach. Specifically, a main terms logistic regression model was fit to the data adjusting for journal, primary disease subject, geographic region, and year, which was included as a linear term. A multi-degree-of-freedom Wald test was used to test whether all regression coefficients associated with each variable were equal to zero. We also summarized the logistic regression model by presenting proportions of LIC-affiliated authorship for each category of each variable standardized to the distribution of study characteristics in our sample. For year, which is modeled as a linear term, we present the standardized proportion every six years. Confidence intervals for the standardized proportions were computed using the nonparametric percentile bootstrap with 500 bootstrap samples. All analyses were completed using R (https://cran.r-project.org/).

## Results

Among the 23,631 articles published in IDSA journals between 1998 and 2017, a total of 3233 were identified as being conducted within a LIC. After applying our inclusion and exclusion criteria, there were 1308 articles remaining which were included in the analysis (Fig 1). A manual audit of 50 randomly selected articles for validation of the Scopus search criteria and data

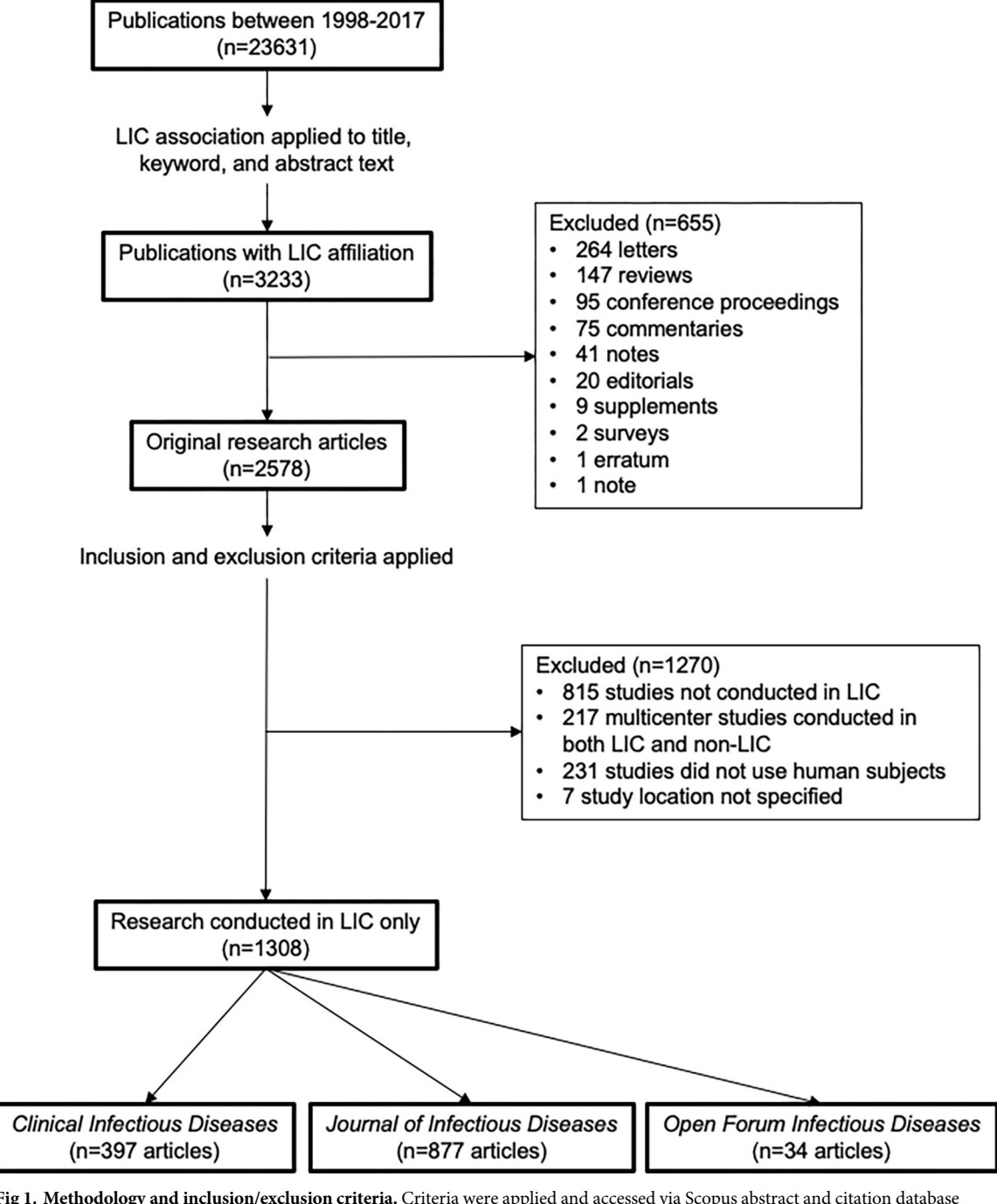

**Fig 1. Methodology and inclusion/exclusion criteria.** Criteria were applied and accessed via Scopus abstract and citation database search strategy. Abbreviations: LIC, low-income country.

extraction demonstrated 96% accuracy (number of articles with correct study location and author affiliations/number of articles audited) in capturing the LIC setting for the study, 96% accuracy of first authorship affiliation and 98% accuracy of last authorship affiliation. Errors in determining LIC setting were due to countries named in the abstract that were not the setting of the study. The error in identification of authorship affiliation for dual-affiliated authors was a result of not capturing their non-LIC affiliation.

## Authorship trends

The total number of publications conducted in LICs that were published in IDSA journals increased by an average of 2.9 articles per year. There was no significant increase in the number of LIC articles by year when *Open Forum Infectious Diseases* was established in 2014. While 1998 was atypical for only containing five LIC publications, the remaining 19 years ranged from 41 articles in 1999 to 60 articles in 2017 with a maximum of 88 articles in 2009. LIC-based articles comprised 5.0% of articles published between 1998–2007 and 6.3% of articles published between 2008–2017.

Over the 20-year period, 42% of LIC articles had a first author with a LIC affiliation while 29% had a last author with a LIC affiliation. Fifty percent of the articles had either a first or last LIC-affiliated author. The majority of LIC first author (95%) and last author (94%) affiliations were concordant with a country where the research was conducted.

The absolute number of articles published by LIC-affiliated authors increased marginally over time. LIC first and last authors had an average rate of increase of 0.84 and 0.42 articles per year, respectively. Considering the increase in number of LIC articles published overall, the proportion of articles with LIC first and/or last authorship decreased over time (Fig 2). As seen in Fig 3, the proportion of first or last authors from LICs when adjusted for journal, study

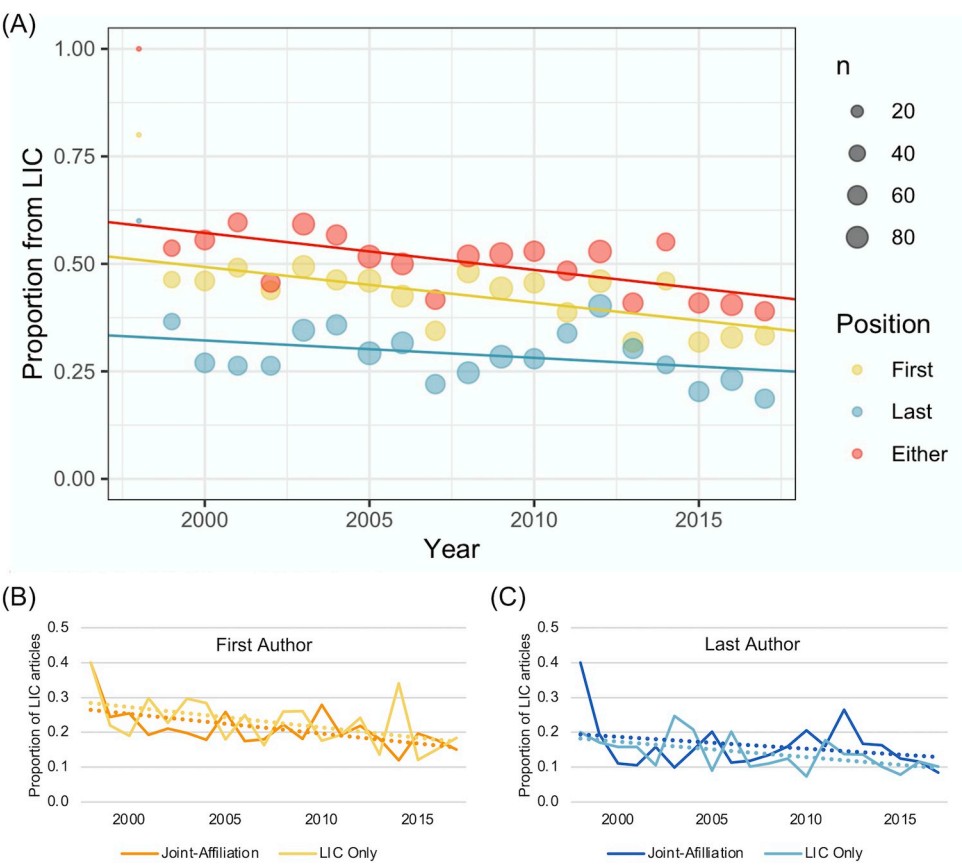

**Fig 2. Proportion of LIC-affiliated first and last authorship over time.** (A) Overall proportion of LIC-affiliated first, last and first or last authorship. Number of articles per year is represented by the size of the corresponding circle. (B) Proportions of LIC-affiliated and jointly-affiliated first authors. (C) Proportions of LIC-affiliated and jointly-affiliated last authors. Abbreviations: LIC, low-income country.

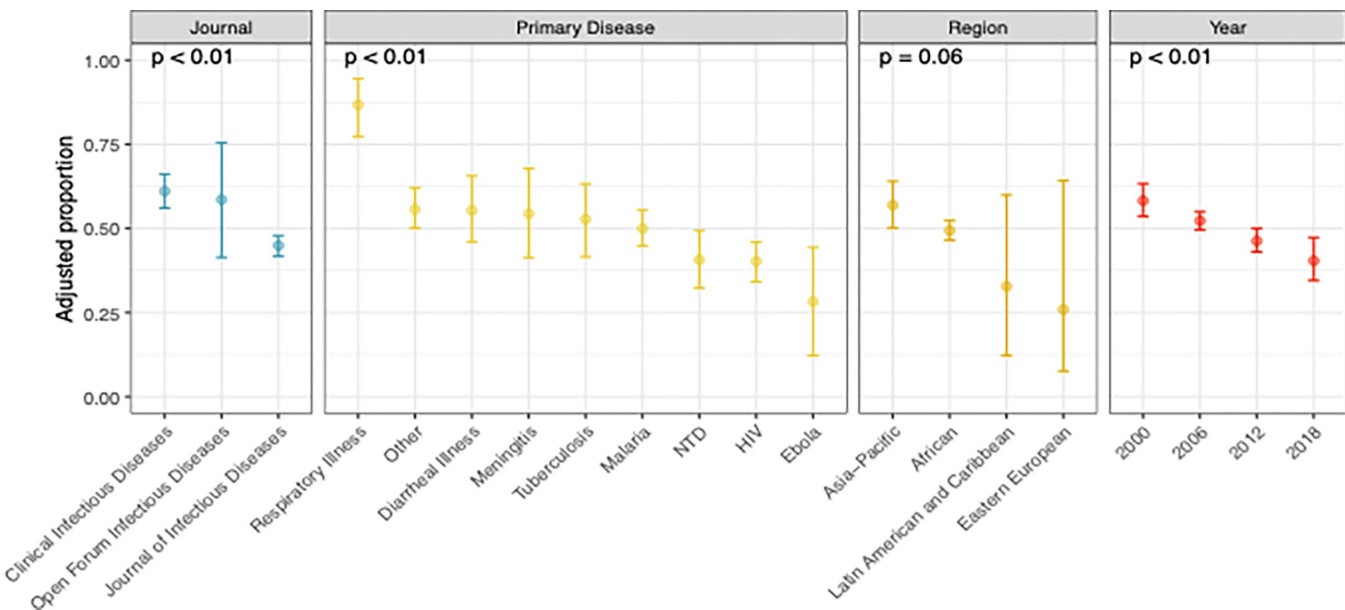

**Fig 3. Adjusted proportion of first or last authors affiliated with low-income countries.** Displayed as adjusted for year, journal, study topic and geographic region. Abbreviations: NTD, neglected tropical diseases; HIV, human immunodeficiency virus.

topic and geographic location was significantly lower between 2013–2017 compared to 1998–2002 (0.40 [CI 0.34–0.46] vs. 0.58 [CI 0.53–0.63], p<0.01).

Dual affiliation by authors with both a LIC and non-LIC affiliation was reported by 48% of LIC first authors and 52% of LIC last authors. Review of the non-LIC affiliations for these authors found that 99% of first authors and 100% of last authors were jointly affiliated with high income countries, most frequently the United States and United Kingdom. The same decreasing trend in proportion of first and last authorship was seen regardless of whether the LIC-affiliated author was dually-affiliated or not (Fig 2B and 2C).

It is important to note that the number of countries that qualified as low income decreased from 63 countries in 1998 to 34 countries in 2017. In a sub-analysis of LIC articles, 73% were affiliated with a LIC that remained as low-income vs. a country that progressed to a higher income classification at some point during the timeframe. The proportion of articles from LICs that remained low-income increased over time (31% in 1998–2006 vs. 41% in 2008–2017) while proportion of articles from countries that changed income classification group decreased (22% in 1998–2006 vs. 6% in 2008–2017) out of all articles published.

## Geographic distribution

Fig 4 contains the distribution of LIC publications and proportion of LIC-affiliated authorship by geographic location. The majority of LIC publications (90%) were conducted in the African continent with 49% of all LIC publications based in East Africa. As outlined in Fig 3, the proportion of LIC first or last authorship was highest in Asia-Pacific region and lowest in Eastern Europe although the low number of articles from Latin America and Eastern Europe resulted in wide confidence intervals for the adjusted proportion of LIC publications. The most frequent affiliations for non-LIC first and last authors, respectively, were the United States (55% and 52%), United Kingdom (13% and 15%), France (6% and 7%) and other high-income countries (22% for both).

(A)

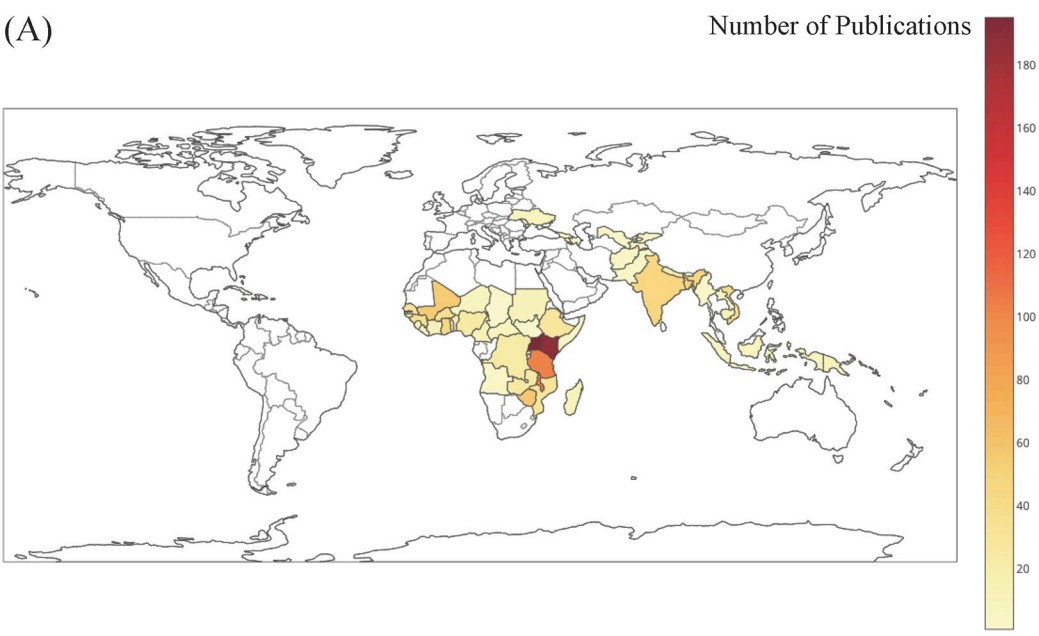

(B)

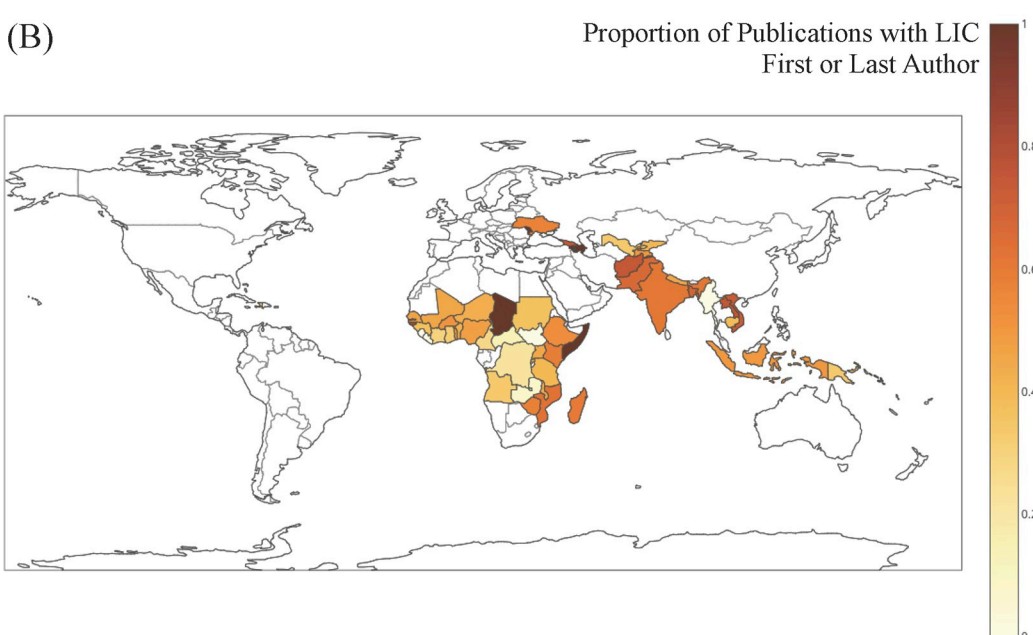

**Fig 4. Geographic distribution of LIC publications.** (A) Total number of publications between 1998–2017 and (B) by proportion of publications with LIC-affiliated first or last authorship. Maps were created using Plotly Chart Studio (Plotly Technologies Inc, released 2015). Abbreviations: LIC, low-income country.

### Study topic

The proportion of LIC publications focused on HIV, malaria, tuberculosis, neglected tropical diseases, diarrheal illness and other infectious diseases remained similar over time (S1 Fig). Within these topics, percentage of LIC first authorship ranged from 36%- 48% and for LIC last authorship ranged from 20%-37%, the results for first or last authorship can be seen in Fig 3.

Publications on Ebola emerged as the predominant topic starting in 2015 and represented the lowest proportion of LIC-affiliated authorship. Respiratory illnesses represented the highest proportion of LIC first and last authors to a significant degree.

## Discussion

We analyzed a subset of primary infectious disease research based in LICs or involving LIC participants published between 1998 and 2017. While the total number of articles authored by LIC investigators increased over time, this is a reflection of the total number of LIC publications increasing over the same period within the three journals investigated. In fact, the proportion of publications authored by LIC-affiliated investigators deceased over time. Our findings indicate more efforts are needed to enhance LIC authorship equity within high-impact infectious disease journals.

High-impact research in LICs has historically been under-represented in the medical literature [18, 19] despite social and ethical arguments for research to be based on relevant global disease burden [20]. A retrospective study by Sumathipala et al. analyzing five leading general medicine journals over a timespan of one year found that only 7.6% of all published articles came from sites outside of Euro-American countries [21]. A similar analysis found that only 3% of research published in the New England Journal of Medicine between 1997 and 2004 was conducted in low-middle income countries (LMICs) [22].

The inequities that result in under-representation of medical research from LICs can also be seen in authorship trends from these areas. While trends in some journals has shown that the vast majority of lead authors were from LICs [23], other analyses in LIC authorship have demonstrated lower rates of LIC authorship, including within the literature for obstetrics [24], tropical medicine [25], and psychiatry [26]. Of the articles published from low-income settings investigated by Sumathipala et al., 69% had one or more authors from Europe or North America and 10% of publications did not include any authors from the country where the research took place [21]. Large-scale metadata on global health research over the past 20 years indicates that while the proportion of non-HIC lead authors has increased over time, the majority of this trend was secondary to increases in authorship from upper-middle- and lower-middle-income countries, with LICs remaining underrepresented [27].

Our study demonstrates a similar distribution of research from LICs within a subset of the infectious disease literature. Between 1998 and 2017, IDSA journals published more research based in LICs with the number of publications with LIC first or last authors increasing over time. Both trends are encouraging and likely reflect the growing international interest in addressing infectious diseases in high disease-burden areas. There have additionally been escalating calls for research in LICs to evolve away from 'parasitic' or 'parachute' models [28] now widely condemned by journal editorial boards [19].

Despite the total number of LIC publications increasing over time, the proportion of first or last authorship by LIC-affiliated investigators decreased. This means that the increase in publications is mostly attributable to non-LIC-affiliated authorship, the vast majority of which are authors from high-income countries. Our results are similar to an investigation of authorship trends in the journal *Lancet Global Health* that revealed while 92% of research publications were conducted in LMICs, only 35% of authors were from LMICs [29]. A study looking at LMIC authorship of randomized control trials on HIV/AIDS, malaria, and tuberculosis between 1990–2013 found that 49.8% of publications had LMIC first authors. While this is a higher proportion than identified by our study, Kelaher et al. found a similar trend that the absolute number of LMIC authors increased but the proportion of LMIC authors decreased over time [30].

Barriers to promoting LIC authorship remain rooted in the inequities that limit research-building capacity and vary greatly between different locations, partnership structures, and types of research. Any engagement in barrier identification and capacity building measures should be led by LIC investigators and communities. Such barriers include, but are not limited to, a paucity of local mentorship [31], lack of research leadership [11], insufficient financial and material resources, delays in regulatory approval or oversight [32], and competing work demands [33]. For research produced by collaboration between high-income and low-income institutions, a predominant model for international health research, inequity between collaborators can create further imbalances. These obstacles include lack of promotion and leadership opportunities for LIC investigators and less emphasis on local health priorities since the high-income partner often provides funding and drives the primary research objectives [34, 35]. Authorship requirements also traditionally include significant contribution to the research described in the manuscript. As most leading journals are published in English, including the three journals analyzed in this study, this excludes authorship qualification for non-English-speaking investigators [9]. Language of publication is further described by Mbaye et al. [36] who found that 49.8% of articles about infectious disease research conducted in Africa featured an African first author and 41.3% an African last author, yet the majority of these authors were affiliated with countries still considered 'English-speaking' like South Africa, Ethiopia, Kenya, and Nigeria. These imbalances extend to other components of the publication process including submission and selection for publication. The high cost of publishing in global health journals widens the access gap for underfunded LIC investigators [37] who also face significant restrictions in access to much of the academic and medical literature. LIC representatives are similarly sparse among journal editorial boards [32]. A review of tropical medicine journals determined that only 5.1% of editorial and board members were affiliated with LICs versus 70.8% from high-income countries [25].

Author co-affiliation with multiple institutions is poorly defined. Authors with affiliations in both high-income and low-income settings may represent important cross-cultural brokers that can assist in translating effective research methods from high-income settings into LICs. But without a universal definition for claiming dual affiliation, there are a wide range of possibilities for what these affiliations mean. This includes LIC investigators who report HIC affiliations with the hopes of benefiting from a positive reputation of a high-income academic partnership. Conversely, there is also a risk that authors may over-report affiliations with LICs to lend credibility and avoid the appearance of parachute research. If the latter theory is implied, we might expect to see an increasing proportion of articles with a dual-affiliated first or last author over time, as compared to the decreasing proportion seen for articles with first or last author with a LIC-only affiliation. However, the same decreasing trend in proportion of first and last authorship was seen for authors with joint affiliations when compared to authors with singular LIC affiliations. This may suggest that the impact of barriers to authorship for LIC investigators is not influenced or improved by co-affiliation with institutions based in high-income countries. Formal criteria for authorship dual affiliation should be explored further to help clarify this issue.

Most publications in this analysis originated from the African continent and in particular East and West Africa. Similarly, prominent representation of research from sub-Saharan Africa has been reported in the global health literature [38]. This may be in part due to multiple long-standing collaborations between high-income academic institutions and African universities [39] combined with the burden of infectious diseases on the local populations. However, these areas did not demonstrate differences in proportion of LIC-affiliated authorship suggesting that increased attention, resources, and number of publications does not always translate to promotion of local investigators. A similar analysis of academic

partnerships within sub-Saharan Africa showed that LIC authorship representation reduced significantly in articles published in collaboration with HICs, in particular when collaborating with high-ranking universities from the United States [40].

Specific diseases targeted by international directives like the UN Millennium Development Goals such as HIV, tuberculosis, and malaria comprised a steady proportion of publications over time. General disease classifications that still contribute significantly to morbidity and mortality in low-income settings such as meningitis, infectious diarrhea and respiratory infections fluctuated expectedly with events such as the Middle East Respiratory Syndrome (MERS) outbreak in 2012 and sporadic meningitis outbreaks in the 2010s. These groups did have higher proportions of LIC-affiliated authorship, although not statistically significant, when compared to HIV, tuberculosis and malaria. This poses interesting questions around the impact of research topic on authorship equity including whether broadly endemic diseases like HIV and tuberculosis, which incur multinational research funding and collaborations, have differing complexities to the promotion of LIC investigator development compared to local disease outbreaks. However, despite being a localized epidemic, literature on Ebola had the lowest proportion of LIC-affiliated authorship suggesting that emergency infectious disease responses are a particularly complex research environment for LIC investigators with complicating factors that include increased clinical demands for local providers, the need for rapid dissemination of research findings, and politicized highly-resourced multiorganizational research collaborations [41]. This is of particular importance in the context of the current global COVID-19 pandemic. There has not only been an unprecedented acceleration of scientific research but also a growing spotlight on inequities of healthcare provision in low-income settings [42, 43] and attention to addressing these disparities through collaborative global partnerships [44]. Data on COVID-19 research conducted in Africa already displays problematic authorship patterns with less than half of articles featuring an African first or last author and 20% of articles with no African authors included [45].

Only 5.5% of articles that were both published in the three journals analyzed and met inclusion criteria were conducted in LIC settings. These journals are not wholly representative of infectious disease research in LICs more generally. The focus on few journals for our analysis acknowledges there is a degree of publication bias within our results. These journals represent only a fraction of infectious disease research published and do not allow for investigation of publication trends across a wider range of journal impact factors. For example, in their analysis of authorship in sub-Saharan Africa, Hedt-Gauthier et al. found that over half (60.6%) of articles published in journals without impact factors had local authorship representation compared to only 4.8% of papers published in journals with impact factors over 10 [40]. A similar analysis within infectious disease literature and broader geographic focus would be of interest and relevant to our findings. While the three IDSA journals have an international focus, this may be limited to specific geographic areas and would benefit from comparison to publications in other journals. The representation of geographic areas may have been impacted by the 20-year period selected for analysis. The same phenomena that contribute to episodic increases in disease topics also applies to geographic locations, for example a higher proportion of research from West Africa in 2015–2017 as a result of the Ebola epidemic. Additionally, the preliminary analysis for this paper was conducted in 2018 and the end date of 2017 does not investigate more recent authorship trends. Due to the nature of abstract extraction there were errors identified in the final data set although these were minor. We were unable to analyze other characteristics that would be of interest, such as author gender or funding source of the research, due to limitations in the information provided by Scopus abstract and citation database. While dual-affiliated authors with both LIC and non-LIC affiliations were included as LIC-affiliated authors this likely resulted in an over-estimation due to this being a poorly

defined variable in the absence of clear guidelines. An additional limitation is that authorship is an imperfect marker for research building capacity. International collaborations have included authors with vague or variable contribution to the research [46]. Although universal authorship guidelines have been proposed by the International Committee of Medical Journal Editors, awareness of these guidelines is low in LICs with frequent guest authorship and manuscript recycling [47].

Despite these limitations, this study demonstrates inherent inequities in authorship within infectious disease research conducted in LICs. Potential mechanisms to address these issues by a top-down approach include changing the promotional scheme for high-income academic investigators to off-load the emphasis of promotion based on the number of first or last author publications. Instead, promotion could be considered for those investigators who provide exceptional mentorship for low-income investigators or be measured as a product of mentee publication. Another method is prioritizing LIC investigators in the selection process for international research funding and encouraging, or requiring by LIC institutional policy, more equitable partnerships when negotiating new collaborations. An example of this includes the Fogarty International Center at the National Institute of Health which require grant recipient organizations and principle investigators to be based in low-income settings [48]. Interventions from a more bottom-up approach include education and training on manuscript preparation, statistics, scientific writing, and other relevant topics to be included in the framework of academic collaborations between high- and low-income settings in such a way that does not place additional burden on the LIC partner.

This study demonstrates that the proportion of infectious disease LIC publications by local authors has decreased over time despite an increase in the number of LIC publications over a 20-year timeframe. These trends demonstrate that significant barriers still exist which exclude LIC investigators from publishing within a subset of the infectious disease literature despite the overwhelming burden of morbidity and mortality posed by infectious diseases within low-income settings. Active efforts are needed to increase attention and further investigation promoting LIC authorship.

## Supporting information

**S1 Fig. Proportion of low-income country articles by study disease topic over time.**
(TIFF)

**S1 Data. Complete and stratified data sets used in the analysis.**
(XLSX)

## Author Contributions

**Conceptualization:** Chelsea E. Modlin, Lance Waller, Kimberly R. Powell, Russell R. Kempker.

**Data curation:** Qiao Deng, Kimberly R. Powell.

**Formal analysis:** Chelsea E. Modlin, Qiao Deng, David Benkeser, Russell R. Kempker.

**Investigation:** Chelsea E. Modlin.

**Methodology:** Chelsea E. Modlin, Qiao Deng, David Benkeser, Lance Waller, Kimberly R. Powell, Russell R. Kempker.

**Supervision:** Russell R. Kempker.

**Visualization:** Chelsea E. Modlin, Qiao Deng, David Benkeser.

**Writing – original draft:** Chelsea E. Modlin, Qiao Deng.

**Writing – review & editing:** Chelsea E. Modlin, Qiao Deng, David Benkeser, Yimtubezinash Woldeamanuel Mulate, Abraham Aseffa, Lance Waller, Kimberly R. Powell, Russell R. Kempker.

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
