## [Decision Letter · Decision Letter 0]

10 Jan 2022

PGPH-D-21-01066

Authorship Trends for Infectious Disease Research Conducted in Low-Income Countries

Dear Dr. Modlin,

Thank you for submitting your manuscript to PLOS Global Public Health. After careful consideration, we feel that it has merit but does not fully meet PLOS Global Public Health’s publication criteria as it currently stands. Therefore, we invite you to submit a revised version of the manuscript that addresses the points raised during the review process.

We look forward to receiving your revised manuscript.

Kind regards,

Bethany Hedt-Gauthier, PhD

Academic Editor

Journal Requirements:

1. Please amend your Data Availability Statement and indicate where the data may be found (domain name and direct link).

2. Please amend your detailed Financial Disclosure statement. This is published with the article, therefore should be completed in full sentences and contain the exact wording you wish to be published.

ii). State the initials, alongside each funding source, of each author to receive each grant.

iii). State what role the funders took in the study. If the funders had no role in your study, please state: “The funders had no role in study design, data collection and analysis, decision to publish, or preparation of the manuscript.”

3. Please include the Funding Information in the system. This should have the same information of Financial Disclosure statement.

Additional Editor Comments (if provided):

Reviewers' comments:

Reviewer's Responses to Questions

**Comments to the Author**

1. Does this manuscript meet PLOS Global Public Health’s publication criteria? Is the manuscript technically sound, and do the data support the conclusions? The manuscript must describe methodologically and ethically rigorous research with conclusions that are appropriately drawn based on the data presented.

Reviewer #1: Partly

Reviewer #2: Yes

2. Has the statistical analysis been performed appropriately and rigorously?

Reviewer #1: Yes

Reviewer #2: Yes

3. Have the authors made all data underlying the findings in their manuscript fully available (please refer to the Data Availability Statement at the start of the manuscript PDF file)?

Reviewer #1: No

Reviewer #2: Yes

4. Is the manuscript presented in an intelligible fashion and written in standard English?

Reviewer #1: Yes

Reviewer #2: Yes

5. Review Comments to the Author

Reviewer #1: In this manuscript, authors investigate the number original research articles pertaining to low-income countries (LICs) published in three infectious disease journals (Clinical Infectious Diseases, Journal of Infectious Diseases, and Open Forum Infectious Diseases), and the number and proportion of these articles that include LIC first- and/or last- authors. Generally, the manuscript is interesting, clear, and extremely well-written; thank you for the opportunity to review. Comments for authors are below:

General Comments:

-Authors may consider expanding the Introduction and Discussion to include the following relevant/recent publications: https://gh.bmj.com/content/6/1/e003758;
https://gh.bmj.com/content/4/5/e001853.abstract;
https://europepmc.org/article/med/21144255. These publications also cite other topic-specific studies of authorship metrics that authors may or may not find relevant.

-Authors may consider reframing parts of the Introduction and Discussion from the perspective(s) of LIC researchers, including the barriers LIC researchers may face, and the contribution of HIC-based systems of research/dissemination to those barriers. I think there are a few statements that could be interpreted as shifting the onus on LIC researchers/systems rather than acknowledging the role that HIC researchers/systems may have in perpetuating the inequities that the authors describe (I recognize this is likely not the authors' intended meaning). I have highlighted some specific statements below that the authors may consider rewording.

Specific Comments:

-Abstract, page 2, lines 26-27: “…the contribution of LIC investigators as measured by authorship metrics has been poorly studied”. I believe this statement is incorrect (see above publications as a sample of other studies that have investigated authorship metrics); perhaps “…the contribution of LIC investigators as measured by authorship metrics, specifically to infectious diseases research, has not been thoroughly studied” or something similar would be more accurate.

-Introduction, page 3, lines 47-50: “While the predominant burden of infectious diseases falls within LICs, these are the geographic areas that have historically lacked the resources and infrastructure to generate impactful locally-led research (4, 5). Although there has been substantial interest in developing research capacity within LICs (6-8), significant obstacles still exist.” See above comment re Introduction/Discussion from the perspective(s) of LICs. Authors may consider checking whether the articles they cite find that LICs do not "generate impactful locally-led research" (i.e. is it possible that impactful locally-led research is generated, but LICs experience barriers in publishing this in HIC-based journals with high impact factors?). Similarly, have LICs "lacked resources and infrastructure" or have LICs encountered barriers in obtaining funding/building infrastructure? Who, specifically, has expressed "substantial interest in developing research capacity"; researchers from LICs, HICs, or both?

-Methods, page 4, line 65: Authors should define primary research (i.e. does this refer to primary data collection or is it synonymous with original research articles - excludes reviews/commentaries?)

-Methods, page 4, lines 66-68: Authors may consider elaborating on how their sample of these three journals represents infectious disease research more broadly. Are these the top 3 journals in this field? Are these most likely to publish research outside of HICs?

-Methods, page 4, lines 77-78: “results were then explored for associations with LICs based on title, keywords, or abstract text” was this done electronically (i.e. text parsing in statistical software) or manually? If manually, were there two reviewers, and how were disagreements resolved?

-Methods: This journal may require authors to mention that ethics approval was waived/not required (at the editor's discretion).

-Results, page 6, line 121: Authors may consider briefly mentioning the formula for accuracy within the Methods section (i.e. I assumed formula was # articles with LIC affiliation/# articles; is this correct?)

-Results, page 7, line 134: Authors should define the acronym OFID on first use.

-Results, page 8, line 164 (also applies to Methods, page 5, line 85): Did % first/last author increase/decrease over time among only the 73% of articles about countries that remained LIC over time? Generally, I am wondering how addition/subtraction of countries from certain years would have influenced authors' results re trends over time.

-Discussion, page 11, lines 223-225: “This means that the increase in publications is attributable to non-LIC authorship” I think these findings are attributable “to a greater extent” to non-LIC authorship, because authors did also see increase in absolute numbers of publications with LIC first-/last- authorship (yet these increases were not proportional to increase in publications).

-Discussion, page 11, lines 232-252, 311-324: See above comment re Introduction/Discussion from the perspective(s) of LICs. Authors may consider mentioning the role of LIC consultation and collaboration re top down and bottom up solutions. It would be, I think, an oversight for HIC researchers to assume they know what LIC want/need re building capacity and addressing barriers.

-Discussion, page 11, lines 253-267: I really liked the discussion of dual affiliation, and I think that exploration of dual affiliation is a strength of this study.

-Discussion, page 13, lines 279-282: “These groups did have higher proportions of LIC authorship, although not statistically significant, when compared to HIV, tuberculosis and malaria, posing the question of the impact of international attention on local investigator development and authorship.” Authors may consider clarifying this sentence (I didn't understand its meaning).

Reviewer #2: It was my pleasure to review this manuscript relating to authorship trends in infectious disease research conducted in low-income countries. The authors provide a transparent, generally well-written and well-investigated analysis that provides a focused and unique perspective (LIC authorship trends in high-impact ID journals) of a larger, well-recognized issue in global health (limited LIC/LMIC author representation in published research). I enjoyed reading this manuscript and feel that the authors have done an excellent job in terms of their analysis. I was particularly impressed by the thoroughness of the discussion. Overall, notwithstanding the irony of HIC first and senior authors on this paper (which implicitly criticizes the same patter in high-impact ID journals), I recommend publication with some minor tweaks, as noted below. I commend the authors on their excellent work.

Abstract

Minor grammar and syntax errors. Throughout the manuscript, please ensure that commas are added prior to dependent clauses, particularly those that begin with adjectives and adverbs.

Introduction

-Does “infectious diseases” need to be capitalized throughout this paper?

Methods

Search Strategy

-Can the authors explain why the study period of 1998-2017 was chosen (given that it is now 2022)? What about this period (and, specifically, the 2017 cutoff) warrants study and why are more recent articles not included? One might argue that the global health movement has gained considerable steam in the past 5 years, so exclusion of articles beyond 2017 might bias the study.

Inclusion/Exclusion

-Line 85: Consider citing the World Bank County and Lending groups website in this section, as opposed to in the ensuing section, given that it is first mentioned here.

Analysis

-See above about reference #16 (World Bank citation).

Results

-Lines 125-6: “Errors in authorship affiliation missed the non-LIC affiliation for dual-affiliated authors although the LIC affiliation was accurate for these data.” I am very confused by this sentence. Can the authors re-phrase for clarity?

Authorship Trends

-Line 134: The abbreviation “OFID” is used here, but should be defined in the search strategy section, where the full Journal title is used for the first time.

-Lines 164-70: The authors raise an interesting question by (appropriately) mentioning that fewer countries were classified as LIC between the beginning and end of the study period. While the insinuation is present in the current text, I wonder whether the same trend of diminishing representative authorship would be present if the analysis included LMICs and LICs, rather than just LICs alone?

Geographic Distribution

-Figure 4: The maps add a nice touch and are easy to navigate. If appropriate, the authors should cite the program used to construct the maps in the Figure 4 heading.

Discussion

-Lines 268-73: If the authors agree, I would consider a brief mention of overrepresentation of sub-Saharan (and, in particular, East African) research, which seems not to be isolated to Infectious Diseases studies (see: https://pubmed.ncbi.nlm.nih.gov/32959620/).

-Lines 282-5: “Literature on Ebola had the lowest proportion of LIC authorship suggesting that emergency epidemics are a particularly complex research environment for LIC investigators with complicating factors that include urgency and politicized multiorganizational research collaborations.” Can the authors better explain what they mean by “urgency”? I would argue that emergency epidemics, such as Ebola Virus Disease, that create public health emergencies in LMIC settings markedly increase the work burden for local practitioners, leading to very little time/capacity to perform original research. Moreover, established, robust and well-funded research programs, most of which are based in HICs and which often collaborate directly with global health public bodies, such as the WHO, CDC and ECDC, usually have much greater capacity to ramp up scientific investigations on short notice.

-Lines 285-91: The authors raise an important point about LMIC representation during the COVID-19 pandemic. It may also be worthwhile to mention that many efforts are underway to address these inequalities through broad global collaborations. See, for example, https://www.ajtmh.org/view/journals/tpmd/104/3_Suppl/article-p1.xml.

-Lines 297-8: Note that “phenomena” is plural, but the verb “contributes” denotes the singular.

6. PLOS authors have the option to publish the peer review history of their article (what does this mean?). If published, this will include your full peer review and any attached files.

**Do you want your identity to be public for this peer review?** For information about this choice, including consent withdrawal, please see our Privacy Policy.

Reviewer #1: No

Reviewer #2: No

---

## [Decision Letter · Decision Letter 1]

30 Mar 2022

PGPH-D-21-01066R1

Authorship Trends for Infectious Disease Research Conducted in Low-Income Countries

Dear Dr. Modlin,

Thank you for submitting your manuscript to PLOS Global Public Health. After careful consideration, we feel that it has merit but does not fully meet PLOS Global Public Health’s publication criteria as it currently stands. Therefore, we invite you to submit a revised version of the manuscript that addresses the points raised during the review process.

Overall, your resubmission was very responsive. We ask that you take similar care in this resubmission, including in your consideration of the newest feedback from reviewer 3.

We look forward to receiving your revised manuscript.

Kind regards,

Bethany Hedt-Gauthier, PhD

Academic Editor

Journal Requirements:

Additional Editor Comments (if provided):

Reviewers' comments:

Reviewer's Responses to Questions

**Comments to the Author**

1. If the authors have adequately addressed your comments raised in a previous round of review and you feel that this manuscript is now acceptable for publication, you may indicate that here to bypass the “Comments to the Author” section, enter your conflict of interest statement in the “Confidential to Editor” section, and submit your "Accept" recommendation.

Reviewer #2: All comments have been addressed

Reviewer #3: (No Response)

2. Does this manuscript meet PLOS Global Public Health’s publication criteria? Is the manuscript technically sound, and do the data support the conclusions? The manuscript must describe methodologically and ethically rigorous research with conclusions that are appropriately drawn based on the data presented.

Reviewer #2: Yes

Reviewer #3: Partly

3. Has the statistical analysis been performed appropriately and rigorously?

Reviewer #2: Yes

Reviewer #3: Yes

4. Have the authors made all data underlying the findings in their manuscript fully available (please refer to the Data Availability Statement at the start of the manuscript PDF file)?

Reviewer #2: Yes

Reviewer #3: Yes

5. Is the manuscript presented in an intelligible fashion and written in standard English?

Reviewer #2: Yes

Reviewer #3: Yes

6. Review Comments to the Author

Reviewer #2: Kudos to the authors for an excellent revision. Thank you for taking heed of the reviewers' recommendations. My two very minor suggestions for revision (neither of which should preclude acceptance of the article) are as follows:

1) In the limitations, please briefly explain why the cutoff date of 2017 was chosen and the impact that this choice could have had on the analysis.

2) "low-income setting" is used in the Introduction, whereas "resource-limited settings" is used in the Discussion. While the two terms often are used synonymously, they can mean different things to different people. Since the study focus is on LICs, consistency in terminology would be beneficial.

Reviewer #3: We have attached a detailed review as a word document because the editorial manager platform restricts reviews to less than 2000 characters.

7. PLOS authors have the option to publish the peer review history of their article (what does this mean?). If published, this will include your full peer review and any attached files.

**Do you want your identity to be public for this peer review?** For information about this choice, including consent withdrawal, please see our Privacy Policy.

Reviewer #2: No

Reviewer #3: **Yes: **Ulrick Sidney Kanmounye

---

## [Decision Letter · Decision Letter 2]

4 May 2022

Authorship Trends for Infectious Disease Research Conducted in Low-Income Countries

PGPH-D-21-01066R2

Dear Dr. Modlin,

We are pleased to inform you that your manuscript 'Authorship Trends for Infectious Disease Research Conducted in Low-Income Countries' has been provisionally accepted for publication in PLOS Global Public Health.

Best regards,

Bethany Hedt-Gauthier, PhD

Academic Editor

We are pleased to accept your paper, but do require that you change the title in line with the final reviewer's inputs. If you choose not to use their specific recommendation, then please use an alternative that captures the spirit of the recommendation.

Reviewer Comments (if any, and for reference):

Reviewer's Responses to Questions

**Comments to the Author**

1. If the authors have adequately addressed your comments raised in a previous round of review and you feel that this manuscript is now acceptable for publication, you may indicate that here to bypass the “Comments to the Author” section, enter your conflict of interest statement in the “Confidential to Editor” section, and submit your "Accept" recommendation.

Reviewer #3: All comments have been addressed

2. Does this manuscript meet PLOS Global Public Health’s publication criteria? Is the manuscript technically sound, and do the data support the conclusions? The manuscript must describe methodologically and ethically rigorous research with conclusions that are appropriately drawn based on the data presented.

Reviewer #3: Yes

3. Has the statistical analysis been performed appropriately and rigorously?

Reviewer #3: Yes

4. Have the authors made all data underlying the findings in their manuscript fully available (please refer to the Data Availability Statement at the start of the manuscript PDF file)?

Reviewer #3: Yes

5. Is the manuscript presented in an intelligible fashion and written in standard English?

Reviewer #3: Yes

6. Review Comments to the Author

Reviewer #3: Thanks for addressing my comments.

I appreciate how intensive this study was and agree that your work is valuable. However, as it stands, the manuscript title and methodology/aim are not aligned.

Hence, I recommend you modify the manuscript title so it reflects your search i.e., a search prior to 2018 and only of Infectious Disease Society of America affiliated journals. I feel this is critical seeing as this study is published in 2022, post-COVID and the search did not include LMIC-based journals. It is reasonable to expect that authorship trends in LMIC-based journals would be very different from those of HIC based journals. For example, looking at the South African Medical Journal and PanAfrican Medical Journal one can see that the majority of authors are affiliated with LMICs. Same goes for Latin American and Asian journals.

A title like "Authorship Trends in Infectious Disease Society of America Affiliated Journal Articles Published Before 2018 and Conducted in Low-Income Countries" would be better representative of your methodology.

7. PLOS authors have the option to publish the peer review history of their article (what does this mean?). If published, this will include your full peer review and any attached files.

**Do you want your identity to be public for this peer review?** For information about this choice, including consent withdrawal, please see our Privacy Policy.

Reviewer #3: **Yes: **Ulrick Sidney Kanmounye
